# Protective Effect of Green Tea Consumption on Colorectal Cancer Varies by Lifestyle Factors

**DOI:** 10.3390/nu11112612

**Published:** 2019-11-01

**Authors:** Hyejin Kim, Jeonghee Lee, Jae Hwan Oh, Hee Jin Chang, Dae Kyung Sohn, Aesun Shin, Jeongseon Kim

**Affiliations:** 1Department of Cancer Biomedical Science, Graduate School of Cancer Science and Policy, National Cancer Center, 323, Ilsan-ro, Ilsandong-gu, Goyang-si, Gyeonggi-do 10408, Korea; heajene@gmail.com (H.K.); jeonghee@ncc.re.kr (J.L.); 2Center for Colorectal Cancer, National Cancer Center Hospital, National Cancer Center, 323, Ilsan-ro, Ilsandong-gu, Goyang-si, Gyeonggi-do 10408, Korea; jayoh@ncc.re.kr (J.H.O.); heejincmd@ncc.re.kr (H.J.C.); gsgsbal@ncc.re.kr (D.K.S.); 3Department of Preventive Medicine, Seoul National University College of Medicine, 103, Deahak-ro, Jongno-gu, Seoul 03080, Korea; shinaesun@snu.ac.kr

**Keywords:** cancer, colorectal cancer, case-control study, green tea, lifestyle factors

## Abstract

The inconsistent findings regarding green tea intake and colorectal cancer (CRC) risk in several epidemiological studies might result from variations in lifestyle factors. Therefore, we examined whether increased green tea intake was associated with a decreased risk of CRC and how the risk of CRC was altered by the protective effect of green tea consumption and five health-related factors. A case-control study including 2742 participants (922 cases and 1820 controls) was conducted in Korea. Green tea consumption was assessed using a semiquantitative food frequency questionnaire. The risk of CRC was approximately 40% less in the participants in the highest green tea intake tertile than in participants in the lowest green tea intake tertile. Of the five lifestyle factors examined, high body mass index and physical inactivity were independent risk factors for CRC. Regarding the interactions between tea consumption and lifestyle factors, high green tea consumption was associated with a decreased risk of CRC, with or without considering lifestyle factors. However, moderate green tea consumption increased the risk of CRC among ever-smokers, ever-drinkers and the high-inflammatory diet group. Increased consumption of green tea might be helpful to reduce the risk of CRC in those with an unhealthy lifestyle.

## 1. Introduction

A rapid increase in colorectal cancer (CRC) incidence has been observed worldwide; CRC rates nearly doubled between 2002 and 2018 due to a shift in lifestyles and diet [1,2,3,4]. Notably, CRC is the second most prevalent cancer in Asian countries, such as South Korea [5,6]. Regarding diet, beverage intake, including tea, increased threefold over a decade in Korea [7]. Among the variety of teas, green tea may have beneficial effects on CRC due to its polyphenol content [8,9]. Moreover, green tea may influence multiple stages of carcinogenesis by reducing antioxidant capacity and preventing inflammation [10,11]. Although experimental studies have demonstrated that green tea strongly contributes to a decreased risk of CRC, previous epidemiological studies have reported inconsistent results [12,13,14]. A recent meta-analysis of 12 case-control studies and 17 cohort studies with 1,642,007 participants identified an inverse association between the risk of CRC and green tea consumption [12]; however, there were no such associations in a Chinese prospective cohort study [13] and an Australian case-control study [14]. Furthermore, the evidence linking CRC with tea consumption was classified as limited [15].

Both green tea intake and lifestyle factors likely have essential roles in colorectal carcinogenesis. A number of health-related factors, such as body mass index (BMI), alcohol consumption, physical activity, smoking and the dietary inflammatory index (DII), have been associated with CRC risk [16,17,18]. Although some of the investigated factors were found to be related to green tea consumption [19], no studies have examined the protective effect of green tea intake considering lifestyle factors on CRC risk in South Korea. Furthermore, the variable results of epidemiological studies on the risk of CRC with green tea intake might be the result of numerous lifestyle factors.

Thus, we examined whether green tea consumption was associated with a lower risk of CRC. Additionally, we determined the interactions between CRC and lifestyle factors and how the risk of CRC was altered by the interaction between green tea consumption and lifestyle factors.

## 2. Materials and Methods

### 2.1. Study Population

The details of this case-control study have been reported elsewhere [18,20]. This case-control study was initiated in 2010 at the National Cancer Center (NCC), Korea. Cases were recruited from patients newly diagnosed with CRC between August 2010 and August 2013 at the Center for CRC. Of the 1070 individuals who agreed to participate in the study, two patients were excluded due to improbable energy intake (<500 or ≥4000 kcal/day), and 145 individuals were excluded due to incomplete dietary data. Therefore, 923 patients were ultimately included in this analysis. Controls were recruited from individuals who visited the Center for Cancer Prevention and Detection at the same hospital for health checkups between October 2007 and December 2014. Among the 14,201 healthy participants, 5044 and 120 individuals were excluded due to incomplete dietary data and implausible energy intake (<500 or ≥4000 kcal/day), respectively. For the remaining 9037 participants, two controls per case were frequency-matched by age (within five years) and sex. Subsequently, 27 participants were excluded due to extreme tea consumption (>1600 g/day). Ultimately, 2742 participants were included in the final analysis (922 cases and 1820 controls; Appendix A). All subjects provided written informed consent. The study was approved by the Institutional Review Board (IRB) of the NCC (IRB number: NCCNCS-10-350 and NCC2015-0202).

### 2.2. Assessment of Lifestyle Factors

Participants were asked to complete self-administered questionnaires on demographic and lifestyle characteristics, such as smoking and physical activity. Dietary intake data were collected from all participants using a validated semiquantitative food frequency questionnaire (SQFFQ) method [21] by a well-trained dietitian. Total energy intake and green tea consumption were obtained from the SQFFQ data, and the average frequency of eating and typical portion sizes in a year were calculated using Computer-Aided Nutritional Analysis Program 4.0 software (Can-Pro 4.0) (Can-Pro 4.0; Korea Nutrition Society, Seoul, Korea). Green tea consumption was defined as the amount of water with dried green tea leaves. We calculated the DII using a previously established equation [22,23]. In our study, 35 food and nutrient components were included to calculate the DII: carbohydrates, protein, fat, fiber, saturated fatty acids, polyunsaturated fatty acids, monounsaturated fatty acids, *n*-3 fatty acids, *n*-6 fatty acids, cholesterol, thiamin, riboflavin, niacin, vitamin B6, vitamin B12, vitamin C, folic acid, vitamin A, vitamin D, vitamin E, β-carotene, iron, magnesium, selenium, zinc, garlic, ginger, onion, pepper, anthocyanidin, flavan-3-ol, flavone, flavanol, flavanone and isoflavone.

### 2.3. Lifestyle Factors

Lifestyle factors, such as prior BMI, smoking (current-, former- or never-smoker), alcohol consumption (current-, former- or never-drinker), physical activity (yes or no) and the DII, were selected based on previous evidence of risk factors for CRC and their association with CRC according to the data [16,17,18,22]. First, we found that prior BMI (prior two years before diagnosis) was more significantly associated with CRC in our study than current BMI, as shown in our previous work [18]. As a result, prior BMI was used as an index for obesity. Second, physical activity was defined as regular physical exertion. Third, alcohol consumption and smoking were chosen and classified as ever or never based on previous studies [16,17]. Fourth, the DII was selected as the criterion for a healthy diet based on our former work showing a positive association between DII and the risk of CRC [18,22], and participants were divided into two groups depending on the median intake value of the controls.

### 2.4. Statistical Analysis

The differences in age and dietary intake (continuous variables) between cases and controls were analyzed using Student’s t-test. Other categorical demographic and lifestyle factors were tested using the chi-squared test.

The associations of green tea consumption with CRC were assessed using logistic regression models adjusted for first-degree family history of CRC (yes or no), education (<12 years or ≥12 years) and lifestyle factors, including prior BMI (≥25 kg/m^2^ or <25 kg/m^2^), regular exercise (yes or no), smoking (ever-smoker or never-smoker), alcohol consumption (ever-drinker or never-drinker) and DII (high or low) according to anatomical site (i.e., colon and rectal cancer). Green tea consumption was divided into tertiles depending on the distribution in the control group, with the lowest tertile considered as a reference for all analyses. Trends were tested with the median values of green tea consumption in each tertile group as continuous variables. Regarding the associations between lifestyle factors and the risk of CRC, multivariable models were used to consider adjusting for first-degree family history of CRC, education and green tea intake, and additionally adjusting for the other lifestyle factors as needed, stratified by anatomical site. Furthermore, multinomial logistic regression analyses were conducted to identify the interactions between lifestyle factors and green tea consumption in relation to CRC risk. The interactions between each health-related factor and green tea intake were analyzed using a likelihood ratio test, with the model containing main effects and covariates, and they were also stratified by anatomical site. To further clarify the risk of CRC considering health-related factors, the association between green tea consumption and CRC risk was stratified by smoking, alcohol consumption and DII values, which may present some interactions in the analysis.

All statistical analyses were performed at the 5% significance level using the SAS statistical package (SAS 9.4; SAS Institute, Cary, NC, USA).

## 3. Results

### 3.1. General Characteristics

The distribution of the general characteristics of the cases and controls is shown in Table 1. Significant differences were observed depending on the group; the case group was more likely to have a lower prior BMI (in the prior two years, *p* = 0.048), a higher family history of CRC (*p* < 0.001), a lower education level (*p* < 0.001) and to exercise less regularly (*p* < 0.001) than the control group. However, no differences were found in age, sex, BMI, alcohol consumption or smoking.

The mean daily energy intake was 2027.6 kcal/day for the case group and 1701.5 kcal/day for the control group (*p* < 0.001). On average, the case group consumed less green tea (*p* < 0.001) and had a higher DII (*p* = 0.001) than the control group.

### 3.2. Associations Between Green Tea Intake and Risk of CRC by Anatomical Site

The association between CRC and green tea intake according to anatomical site was examined (Table 2). The number of patients with CRC, colon cancer and rectal cancer was 922, 462 and 444 participants, respectively. Those in the highest tertile had a lower risk of CRC than those in the lowest tertile (odds ratio (OR) 0.59; 95% confidence interval (95% CI) 0.46–0.76; *p* for trend <0.001) in the multivariate model. After stratifying by anatomical site, these associations were slightly stronger among patients with rectal cancer than among those with colon cancer.

### 3.3. Associations Between Lifestyle Factors and Risk of CRC by Anatomical Site

Table 3 presents the association between lifestyle factors and the risk of CRC. Among the examined factors, a low prior BMI (OR 0.78; 95% CI 0.63–0.97) and high level of physical activity (OR 0.47; 95% CI 0.38–0.57) were associated with a reduced risk of CRC in the multivariate model. In the stratified analysis by anatomical site, a negative association with CRC risk was observed for low prior BMI (OR 0.60; 95% CI 0.46–0.78) and more regular exercise (OR 0.51; 95% CI 0.39–0.66) among colon cancer patients and for a high level of physical activity among rectal cancer patients (OR 0.43; 95% CI 0.33–0.56).

### 3.4. Protective Effect of Green Tea Consumption and Health-Related Factors on CRC

The protective effect of green tea consumption and modifiable lifestyle factors on the risk of CRC is shown in Table 4. Different and obscure risk patterns in relation to green tea and lifestyle factors were revealed. The association between green tea consumption and the risk of CRC, colon cancer and rectal cancer, further stratified by tobacco smoking, alcohol consumption and the DII, are shown in Figure 1, Figure 2 and Figure 3, respectively. Moderate green tea consumption increased the risk of CRC in ever-smokers, ever-drinkers and those with a high DII, but high tea consumption consistently decreased the risk of CRC. In never-smokers, never-drinkers and those with a low DII, high consumption had a protective effect, but moderate green tea consumption had no effect, and these protective effects seemed to be more relevant for rectal cancer than for colon cancer.

## 4. Discussion

In this case-control study, green tea consumption and lifestyle factors contributed to the risk of CRC both independently and mutually. Overall, high green tea consumption, with or without considering lifestyle factors, decreased the risk of CRC. However, regarding moderate green tea intake, different risk patterns were observed depending on healthy and unhealthy lifestyles. In particular, an increased risk of CRC was shown among ever-smokers, ever-drinkers and the high-inflammatory diet group. The protective effect of green tea on the risk of CRC varied according to lifestyle factors and was likely to be stronger in rectal cancer patients than in colon cancer patients.

The effects of green tea consumption on CRC risk remain inconsistent. In the present study, we found that high green tea intake was associated with a decreased risk of overall CRC, as shown in previous studies [12,19,24]. Additionally, in agreement with previous evidence [25], the protective effects of green tea were likely substantial among those with an unhealthy lifestyle in this study. Recently, a meta-analysis of 29 studies from Asia, Europe, Australia and America with 1,642,007 participants examining the association between the risk of CRC and overall tea intake in stratified analyses revealed an inverse association among green tea consumption (OR 0.87; 95% CI 0.43–0.98), female sex (OR 0.86; 95% CI 0.78–0.94) and rectal cancer (OR 0.91; 95% CI 0.85–0.99) [12]. Green tea extract can impair key transcription factors in human colon cancer cell lines [26] and reduce gene expression in rectal cancer cells in vitro [11]. However, a borderline modest positive association between tea consumption (not herbal) and colon cancer was reported in a pooled analysis of 13 prospective cohort studies conducted in North America and Europe relative risk (RR) for an increase of 250 g/day, 1.04; 95% CI 1.00–1.07) [27]. No such associations between green tea consumption and CRC risk by anatomical location were observed after adjusting for potential confounders in a Chinese prospective cohort study [13] or an Australian case-control study [14]. However, these studies were conducted among participants who mostly did not consume green tea (nearly 85% of the participants were not green tea drinkers) [14] or those with an already healthy lifestyle, such as women who do not smoke or drink alcohol [13], suggesting that in addition to high green tea consumption, lifestyle factors could have a substantial principal role in CRC incidence.

High BMI and physical inactivity were independent risk factors for CRC in this study; physical activity was the strongest among the investigated health-related factors associated with CRC by anatomical site. Several previous studies have found that a variety of health-related factors, such as elevated fasting glucose, high total cholesterol, no use of postmenopausal hormones, no use of aspirin, no cancer screening, and being tall were associated with an increased risk of CRC [19,28,29,30]. In the Physician’s Health Study, age, smoking and daily alcohol use were risk factors for colon and rectal cancer [29]. In the Shanghai Women’s Health Study, age, BMI, waist-to-hip ratio, physical activity, red meat intake and vegetable and fruit intake were identified as risk factors for overall CRC [19]. Interestingly, Yang et al. [19] reported that interactions between green tea consumption and lifestyle factors were observed but were not significant; the risk of CRC among regular green tea drinkers was substantially reduced compared with that among non-drinkers, but only in participants who had an unhealthy lifestyle (i.e., stratified by physical activity as yes, RR 0.64; 95% CI 0.38–1.07, or no, RR 0.52; 95% CI 0.31–0.88; *p* for interaction = 0.57). In the present study, we observed significant interactions of green tea consumption with tobacco smoking, alcohol consumption and a high DII (data not shown). Moreover, high green tea intake was associated with a decreased risk of CRC, but an inconsistent risk depending on lifestyle factors was found in the group of moderate green tea intake, similar to previous findings [25,31].

Although a dose response association between green tea intake and the risk of CRC was examined in previous epidemiological studies [12,32,33], we found different risk patterns for CRC and the amount of green tea consumption depending on smoking and drinking status and a high-inflammatory diet. High green tea consumption decreased the risk of CRC, but moderate green tea consumption increased the risk of CRC in ever-smokers, ever-drinkers and the high-DII group. In never-smokers, never-drinkers and the low-DII group, high green tea consumption also had a protective effect, but moderate consumption had no effect. Similarly, a Chinese hospital-based case-control study [34] reported inconsistent results; tea consumption roughly decreased oral cancer risk. However, in the analysis of the joint effects of smoking, alcohol consumption and tea consumption considering five categories (zero and quartile one to four), the highest cancer risk was shown in the middle category compared to the other categories. There is no evidence of the protective effect of green tea intake interacting with tobacco smoking and alcohol intake on CRC. At most, three likely explanations for this finding could be as follows. First, each lifestyle factor (smoking, alcohol consumption and a high-inflammatory diet) might attenuate the protective effect of green tea on CRC. It is clear that the examined factors are known to be associated with predispositions to various diseases, including cancer [22,35,36]. Furthermore, the synergistic effect between alcohol and tobacco in cancer has been reported [34,35]. In this study, nearly 70% of the ever-drinkers were ever-smokers (data not shown), suggesting that the synergistic effect possibly already modified the true association between health-related factors and CRC. Second, the temperature of tea, age at tea-drinking initiation, concentration of tea consumed and duration of tea consumption, which are known as modifiable factors on cancer risk [34], were not considered in this study. Third, the amount of green tea consumption in the second tertile (maximum, 21.28 g/day; median, 5.08 g/day) might be insufficient to identify an association with the risk of CRC. Based on the evidence regarding the concentration of catechins in green tea [37], it could be speculated that nearly 4.26 mg to 8.51 mg of catechins were consumed by the patients in the second tertile of green tea consumption. However, Bettuzzi et al. [38] showed that the efficacy of green tea on reducing premalignant prostate cancer lesions was due to catechins at 600 mg/day. Future studies are needed to clarify the dose response association between green tea consumption and CRC.

The underlying mechanism of the protective effect of green tea considering lifestyle factors on CRC requires further investigation, but green tea consumption and lifestyle factors may contribute to the risk of CRC both independently and mutually. Green tea might modify the colorectal carcinogenic process through several intracellular and extracellular processes, such as antioxidant activity, inflammation reduction, gut microbiota alterations, enzymatic inhibition in lipid or glucose metabolism and epigenetic changes [10,11,26,39]. Recent animal studies have demonstrated that green tea extract can inhibit the occurrence and formation of precancerous lesions in the colon [8,40]. In particular, the protective effect of green tea was found to be more effective in a group of mice fed a Western diet, which was high in fat and simple sugars, than in a group of mice fed a control diet [40]. Furthermore, experimental studies have shown that a green tea polyphenol, epigallocatechin-3-gallate (EGCG), suppresses the activity of colorectal cancer stem cells [9,29,41], thereby leading to the regulation of cellular proliferation, differentiation and apoptosis, especially through the Wnt signaling pathway [9,39,42]. Additionally, EGCG modulates a number of mRNAs and proteins in colorectal carcinogenesis [26,42]. However, the protective effect of green tea on CRC risk could be modified by environmental factors. Cigarettes contain more than 60 carcinogenic chemicals, and the risk of CRC may differ based on an individual’s genetic susceptibility to cigarette smoke [43]. With respect to alcohol consumption and an unhealthy diet [41,44], the interaction of environmental factors with gene polymorphisms could affect the association with the risk of CRC. Moreover, the anatomical site may affect the protective effect of green tea considering lifestyle factors. In the present study, a similar pattern of the protective effect of green tea and lifestyle factors on CRC was found for rectal cancer but not colon cancer. Because the colon and rectum differ in their physiologically multifunctional processes, such as bile acid metabolism, enzyme activity and microbiota composition, environmental factors and green tea intake may have different effects on colorectal carcinogenesis due to the anatomical site [42,45,46]. In particular, a lower level of Bifidobacterium, which is known to protect the gut microbial environment, was found in patients with rectal neoplasms [47]. Moreover, a case-control study [24] and a meta-analysis [12] also reported inverse associations of green tea and total tea intake, respectively, with CRC risk that were confined to rectal cancer. However, the evidence is limited, warranting further studies on any related mechanisms.

The protective effect of high green tea consumption on the incidence of CRC might be helpful for individuals with unhealthy lifestyle habits, such as alcohol or tobacco use. Additionally, public health interventions to promote both an increased intake of green tea and adherence to a healthy lifestyle to reduce the risk of CRC should be considered. Nevertheless, this study had several limitations that should be noted. First, on the basis of the case-control condition, this study might have potential information and selection bias; the controls may have been more health-conscious than the cases. This factor may result in an overestimated association between green tea consumption and CRC. Second, residual confounding should be mentioned; it likely exists to distort the true association between green tea intake and the risk of CRC. Third, the information on green tea might not be sufficient to evaluate the actual amount of green tea intake because our database did not have any green tea extract information, and the levels of bioactive compounds in green tea vary by season, climate and tea processing conditions [37]. Fourth, because this study included only the Korean population, this result cannot be generalized; thus, it should be interpreted with caution.

## 5. Conclusions

In conclusion, both high green tea intake and a healthy lifestyle were independently associated with a reduced risk of CRC. The protective effect of green tea consumption considering lifestyle factors on CRC was evidently shown, especially in the group with the highest green tea intake. These findings could provide a new understanding of the beneficial role of green tea consumption in the etiology of CRC and a new strategy for the prevention of CRC depending on smoking and drinking status.

## Figures and Tables

**Figure 1 nutrients-11-02612-f001:**
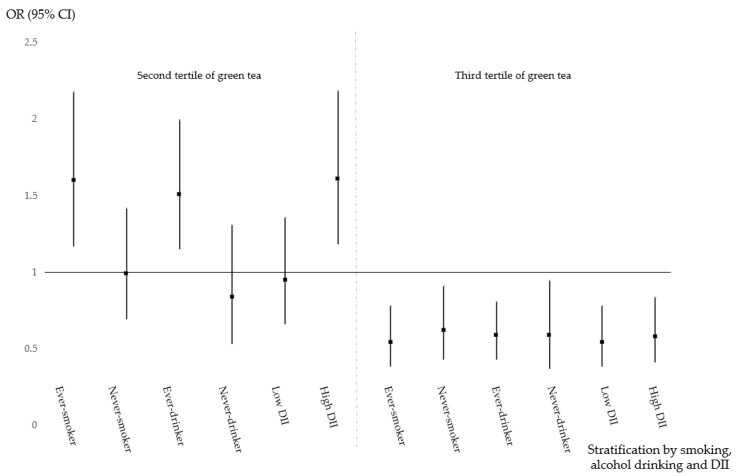
Green tea consumption and the risk of CRC stratified by demographic characteristics.

**Figure 2 nutrients-11-02612-f002:**
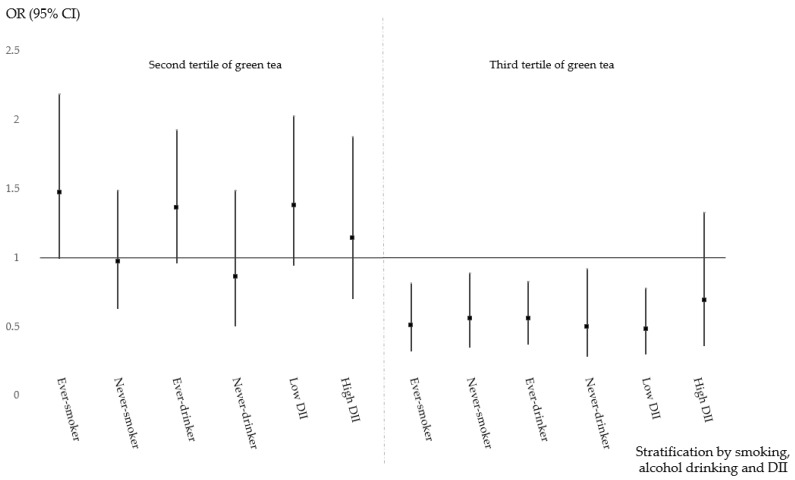
Green tea consumption and the risk of colon cancer stratified by demographic characteristics.

**Figure 3 nutrients-11-02612-f003:**
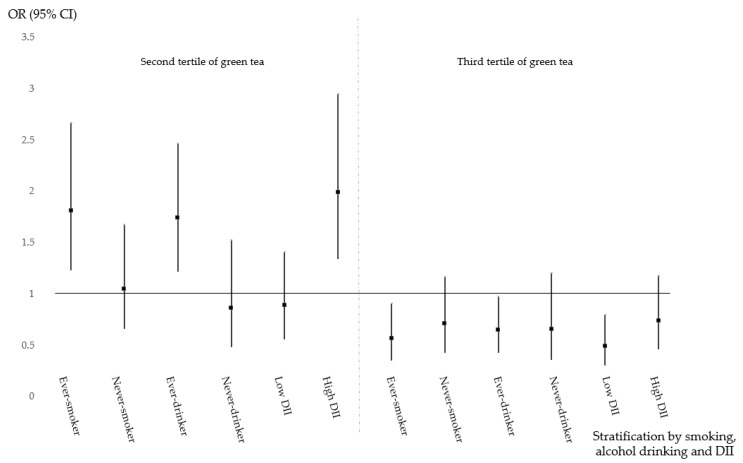
Green tea consumption and the risk of rectal cancer stratified by demographic characteristics.

**Table 1 nutrients-11-02612-t001:** General characteristics of the study population.

Characteristics	Controls (*n* = 1820)	Cases (*n* = 922)	*p*-Value
Age (years)	56.1 ± 9.1	56.6 ± 9.7	0.18 ^1^
Sex (men)	1235 (67.9)	624 (67.7)	0.93 ^2^
BMI (kg/m^2^)			
<25	1210 (66.5)	639 (69.3)	0.14 ^2^
≥25	610 (33.5)	283 (30.7)	
Prior BMI (kg/m^2^) ^3^			
<25	1239 (68.1)	593 (64.3)	0.048 ^2^
≥25	581 (31.9)	329 (35.7)	
Family history of CRC (yes) ^4^	96 (5.3)	86 (9.3)	<0.001 ^2^
Education (years)			
<12	852 (47.9)	690 (74.8)	<0.001 ^2^
≥12	926 (52.1)	232 (25.2)	
Regular exercise (yes)	1030 (58.1)	310 (33.6)	<0.001 ^2^
Alcohol consumption			
Never-drinker	552 (30.3)	279 (30.3)	0.97 ^2^
Ever-drinker	1268 (69.7)	643 (69.7)	
Smoking status			
Never-smoker	801 (44.0)	408 (44.3)	0.91 ^2^
Ever-smoker	1019 (56.0)	514 (55.7)	
Total energy intake (kcal/d)	1701.5 ± 554.6	2027.6 ± 533.0	<0.001 ^2^
DII ^5^	2.7 ± 1.2	2.9 ± 1.3	0.001 ^1^
Green tea consumption (g/d) ^5^	65.8 ± 169.1	22.6 ± 75.7	<0.001 ^1^

BMI, body mass index; CRC, colorectal cancer; DII, dietary inflammatory index. ^1^ Student’s *t*-test. ^2^ Chi-squared test. ^3^ BMI from two years ago. ^4^ First-degree relative. ^5^ DII and green tea consumption were adjusted for total energy intake using the residual method.

**Table 2 nutrients-11-02612-t002:** Associations between green tea intake and CRC risk by anatomical site.

Green Tea Intake (g/d)	No. of Controls	No. of Cases	Crude OR (95% CI)	Multivariate OR (95% CI) ^1^
**CRC**				
T1 (≤0.01)	605	368	1.0 (ref)	1.0 (ref)
T2 (0.02–25.49)	607	378	1.02 (0.85–1.23)	1.29 (1.02–1.63)
T3 (≥25.50)	608	176	0.48 (0.39–0.59)	0.59 (0.46–0.76)
*p* for trend			<0.001	<0.001
**Colon cancer**				
T1 (≤0.01)	605	179	1.0 (ref)	1.0 (ref)
T2 (0.02–21.28)	607	178	0.99 (0.78–1.25)	1.09 (0.81–1.47)
T3 (≥21.29)	608	105	0.51 (0.39–0.66)	0.56 (0.41–0.78)
*p* for trend			<0.001	<0.001
**Rectal cancer**				
T1 (≤0.01)	605	174	1.0 (ref)	1.0 (ref)
T2 (0.02–21.51)	607	176	1.01 (0.79–1.28)	1.35 (1.00–1.82)
T3 (≥21.52)	608	94	0.47 (0.36–0.62)	0.60 (0.43–0.84)
*p* for trend			<0.001	<0.001

OR, odds ratio; T, tertile; 95% CI, 95% confidence interval. ^1^ Adjusted for first-degree family history of CRC, education, prior BMI, physical activity, smoking, alcohol drinking and DII.

**Table 3 nutrients-11-02612-t003:** Associations between modifiable health-related factors and CRC risk by anatomical site.

Categories of Lifestyle Factors	No. of Controls	CRC					Colon Cancer					Rectal Cancer				
		No. of Cases	Crude OR (95% CI)	*p*-Value	Multivariate OR (95% CI) ^1^	*p*-Value	No. of Cases	Crude OR (95% CI)	*p*-Value	Multivariate OR (95% CI) ^1^	*p*-Value	No. of Cases	Crude OR (95% CI)	*p*-Value	Multivariate OR (95% CI) ^1^	*p*-Value
**Prior BMI**																
≥25 kg/m^2^	581	329	1.0 (ref)	0.048	1.0 (ref)	0.02	190	1.0 (ref)	<0.001	1.0 (ref)	<0.001	137	1.0 (ref)	0.66	1.0 (ref)	0.84
<25 kg/m^2^	1239	593	0.85 (0.72–1.00)		0.78 (0.63–0.97)		272	0.67 (0.54–0.83)		0.60 (0.46–0.78)		307	1.05 (0.84–1.32)		0.97 (0.74–1.29)	
**Physical activity**																
No	790	612	1.0 (ref)	<0.001	1.0 (ref)	<0.001	299	1.0 (ref)	<0.001	1.0 (ref)	<0.001	301	1.0 (ref)	<0.001	1.0 (ref)	<0.001
Yes	1030	310	0.39 (0.33–0.46)		0.47 (0.38–0.57)		163	0.42 (0.34–0.52)		0.51 (0.39–0.66)		143	0.36 (0.29–0.45)		0.43 (0.33–0.56)	
**Smoking**																
Ever	1019	514	1.0 (ref)	0.9	1.0 (ref)	0.56	238	1.0 (ref)	0.08	1.0 (ref)	0.07	266	1.0 (ref)	0.14	1.0 (ref)	0.41
Never	801	408	1.01 (0.86–1.18)		1.07 (0.86–1.33)		224	1.20 (0.98–1.47)		1.30 (0.98–1.71)		178	0.85 (0.67–1.05)		0.89 (0.67–1.18)	
**Alcohol drinking**																
Ever	1268	643	1.0 (ref)	0.97	1.0 (ref)	0.08	318	1.0 (ref)	0.73	1.0 (ref)	0.16	315	1.0 (ref)	0.6	1.0 (ref)	0.29
Never	552	279	1.00 (0.84–1.18)		0.81 (0.64–1.03)		144	1.04 (0.83–1.30)		0.81 (0.60–1.09)		129	0.94 (0.75–1.18)		0.85 (0.62–1.15)	
**DII**																
High	682	390	1.0 (ref)	0.012	1.0 (ref)	0.14	189	1.0 (ref)	0.2	1.0 (ref)	0.49	193	1.0 (ref)	0.015	1.0 (ref)	0.20
Low	681	308	0.79 (0.66–0.95)		0.86 (0.70–1.05)		162	0.86 (0.68–1.09)		0.91 (0.71–1.18)		143	0.74 (0.58–0.94)		0.84 (0.65–1.10)	

BMI, body mass index; DII, dietary inflammatory index; OR, odds ratio; 95% CI, 95% confidence interval. ^1^ Adjusted for first-degree family history of CRC, education and green tea consumption. Mutually adjusted for prior BMI, physical activity, smoking, alcohol drinking and DII, if applicable.

**Table 4 nutrients-11-02612-t004:** Protective effect of green tea consumption and health-related factors on CRC.

Anatomical Sites	Lifestyle Factors	Categories	Green Tea Intake (g/day)			*p-*Value ^1^
			T1 (≤0.01)	T2 (0.02–25.49)	T3 (≥25.50)	
CRC	Prior BMI	≥25 kg/m^2^	1.0 (ref)	1.16 (0.77–1.74)	0.45 (0.28–0.71)	0.36
		<25 kg/m^2^	0.67 (0.48–0.93)	1.17 (0.72–1.92)	1.49 (0.86–2.58)	
	Physical activity	No	1.0 (ref)	1.34 (0.99–1.81)	0.59 (0.43–0.82)	0.92
		Yes	0.48 (0.35–0.67)	0.91 (0.56–1.45)	0.97 (0.58–1.63)	
	Smoking	Ever	1.0 (ref)	1.59 (1.17–2.17)	0.55 (0.38–0.78)	0.041
		Never	1.23 (0.88–1.72)	0.61 (0.38–0.78)	1.12 (0.67–1.87)	
	Alcohol drinking	Ever	1.0 (ref)	1.58 (1.19–2.08)	0.61 (0.45–0.84)	0.024
		Never	1.07 (0.75–1.52)	0.51 (0.31–0.84)	0.89 (0.52–1.53)	
	DII	High (≥2.73)	1.0 (ref)	1.65 (1.21–2.25)	0.60 (0.41–0.85)	0.06
		Low (<2.73)	1.05 (0.76–1.46)	0.58 (0.36–0.92)	0.90 (0.54–1.51)	
Colon cancer	Prior BMI	≥25 kg/m^2^	1.0 (ref)	1.07 (0.66–1.73)	0.48 (0.28–0.81)	0.76
		<25 kg/m^2^	0.53 (0.35–0.79)	1.21 (0.66–2.20)	1.24 (0.63–2.42)	
	Physical activity	No	1.0 (ref)	1.34 (0.93–1.95)	0.58 (0.38–0.88)	0.63
		Yes	0.58 (0.39–0.88)	0.75 (0.41–1.35)	0.85 (0.43–1.65)	
	Smoking	Ever	1.0 (ref)	1.49 (1.00–2.21)	0.52 (0.32–0.84)	0.2
		Never	1.50 (0.99–2.27)	0.63 (0.35–1.12)	1.06 (0.55–2.03)	
	Alcohol drinking	Ever	1.0 (ref)	1.44 (1.01–2.04)	0.59 (0.39–0.88)	0.19
		Never	1.06 (0.69–1.64)	0.56 (0.30–1.04)	0.80 (0.40–1.60)	
	DII	High (≥2.73)	1.0 (ref)	1.45 (0.98–2.14)	0.51 (0.32–0.83)	0.29
		Low (<2.73)	1.02 (0.68–1.55)	0.67 (0.37–1.20)	1.05 (0.54–2.04)	
Rectal cancer	Prior BMI	≥25 kg/m^2^	1.0 (ref)	1.35 (0.80–2.30)	0.38 (0.19–0.74)	0.16
		<25 kg/m^2^	0.80 (0.52–1.25)	1.09 (0.58–2.06)	2.08 (0.96–4.51)	
	Physical activity	No	1.0 (ref)	1.41 (0.98–2.03)	0.61 (0.40–0.93)	0.91
		Yes	0.39 (0.25–0.62)	1.07 (0.57–1.99)	1.16 (0.58–2.34)	
	Smoking	Ever	1.0 (ref)	1.77 (1.21–2.59)	0.57 (0.36–0.90)	0.08
		Never	1.00 (0.65–1.54)	0.60 (0.33–1.10)	1.25 (0.64–2.44)	
	Alcohol drinking	Ever	1.0 (ref)	1.78 (1.25–2.54)	0.67 (0.44–1.00)	0.06
		Never	1.14 (0.73–1.80)	0.47 (0.25–0.91)	0.90 (0.44–1.83)	
	DII	High (≥2.73)	1.0 (ref)	1.99 (1.35–2.94)	0.73 (0.46–1.17)	0.04
		Low (<2.73)	1.20 (0.79–1.83)	0.46 (0.25–0.84)	0.69 (0.35–1.35)	

BMI, body mass index; DII, dietary inflammatory index; T, tertile. Multivariate models were constructed to calculate ORs and 95% CIs for the risk of CRC adjusting for first-degree family history of CRC and education. Models were mutually adjusted for prior BMI, physical activity, smoking, alcohol drinking and DII, if applicable. ^1^ The *p* value was calculated from joint tests

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
