# Peer review of "Protective Effect of Green Tea Consumption on Colorectal Cancer Varies by Lifestyle Factors"

_nutrients, 2019, doi:10.3390/nu11112612_

Round 1

Reviewer 1 Report

The manuscript is well written and the topic is interesting and original. However, some aspects could be improved. Please, consider the following suggestions.

The definition of regular physical exertion is missed. please add. Regarding smoking and alcohol consumption, how did the Authors considered the former consumers? As ever or never? Please, specify the acronyms the first time they appear. For instance DII. I suggest to show the number of participants with colorectal cancer, colon and rectal cancer (Table 1 or in the text). please add the unit of measurement in table 2, 3  and 4) please add in table 2 the model 1 only adjusted for family history of CRC and education (instead of only crude model and fully adjusted). add the p value also in table 3. it would be interesting to estimate the ORs stratifying by sex, considering the growing interest in gender health and medicine. the authors find a stronger association between green tea consumption and rectal cancer. however, the authors commented this results only at line 237. can they further argue this aspect in the discussion section? why the authors stated the following: "the controls may have been more conscious about a healthy body image or healthy lifestyle than the cases.". Please argue. nother limitation is the self-reported information (recall bias), moreover, cases are more prone to record more detailed information compared to control, or might over-estimate the consumption resulting in an over-estimated association between tea consumption and CRC. moreover, cases might modify their lifestyle because of the disease. further, authors included only korean population. it means that their results cannot be generalized to other ethinic group, or it should be done with caution. Please add these considerations.  I would suggest referring to the following publications:

-Rectal Cancer: 20% Risk Reduction Thanks to Dietary Fibre Intake. Systematic Review and Meta-Analysis.

-Green Tea Consumption and Risk of Breast Cancer and Recurrence-A Systematic Review and Meta-Analysis of Observational Studies.

-Is dietary fibre truly protective against colon cancer? A systematic review and meta-analysis.

Reviewer 2 Report

This study examined the association between green tea consumption and its interaction with many lifestyle factors with risk of CRC. The study question is important and the study design is appropriate. However, there are some things that needs clarification or improvement.

In the methods part, it said green tea consumption was divided into tertiles with the lowest tertile considered as reference. It also said that “trends were tested with the median values of green tea consumption in each tertile group as continuous variables. It is not clear to me in which model green tea consumption was treated as categorical variable and in which model it was treated as continuous variable.

As stated by the author, some lifestyle factors might be correlated with green tea assumption. It would be good to see how they were correlated in this study. When the authors examined the association between green tea assumption and CRC risk, they adjusted lifestyle factors (Table 2). However, they did not adjust green tea assumption when they examined the association between lifestyle factors and CRC risk (Table 3).

Was the statement on line 152 (page 6) “the risk of CRC was reduced with increased BMI, less regular exercise, a high DII, ever smoking and ever alcohol consumption” based on the results from Table 3? But I do not see a significant association between smoking, drinking and CRC in Table 3. Table 4 only presented interaction effect without showing any main effect results. Thus I don’t see how Table 4 support that statement either. The sentence "the interactions of CRC xxxxx" does not make sense. Also I would like to know what was known about the relationship between those lifestyle factors and CRC and how the observation in this study fit into the literature. They touched some of it in the discussion, but not in a clear comparative way.

Table 4 and interaction results are hard to interpret. First of all, was the green tea consumption treated as categorical variable or continuous variable? From the odds ratio reported in Table 4, it seems like the moderate tea consumption (T2) increase the risk of CRC in ever smoker, but heavy tea consumption decrease the risk of CRC. In never smoker, moderate green tea consumption has protective effect, but high green tea consumption has no effect. Similar observation was found for smoking by green tea interaction. These observations are kind of counter intuitive and are the main finding of this paper. However, the authors did not spend time to clearly present those results other than saying there was an interaction there (section 3.4). Also I found the discussion does not fit well with the results. Also I don’t see how the results support the conclusion that the increased consumption of green tea might be helpful for those with unhealthy lifestyle.

Table 4 is busy and might be hard for the readers to digest and get the main findings, especially the interaction effect. It might be clearer to present interaction effects using Figures.

I would like to see p values on Table 3 like what was done on Table 2 and 4. Also it seems like Table 2 miss the key variable name on the very left column. 

Round 2

Reviewer 1 Report

I thank the Authors for the efforts done to improve the manuscript. However, I'm not fully satisfied especially regarding point 3 and 4 of my previous suggestions.

Author Response

Responses to Reviewer #1
General comment:
I thank the Authors for the efforts done to improve the manuscript. However, I'm not fully satisfied
especially regarding point 3 and 4 of my previous suggestions.
Authors’ response:
We truly appreciate the reviewer’s time and insightful recommendations for improving the
manuscript. We have considered each comment and suggestion during the revision of the
manuscript. Responses to the 2 comments are provided below.
Specific essential comments:
1. It would be interesting to estimate the ORs stratifying by sex, considering the growing interest in
gender health and medicine. the authors find a stronger association between green tea consumption
and rectal cancer. however, the authors commented this results only at line 237. can they further argue
this aspect in the discussion section?
Authors’ response:
Thank you for these thoughtful comments. We have added two more references related to an
association between green tea consumption and rectal cancer and have included sentences in
Conclusion section as follows.
In the manuscript:
Pages 11, Discussion, lines 265-270:
(Lines 265-270) Because the colon and rectum differ in their physiologically multifunctional
processes, such as bile acid metabolism, enzyme activity and microbiota, environmental factors and
green tea intake may have different effects on colorectal carcinogenesis due to the anatomical site [41,
45, 46]. In particular, a lower level of Bifidobacterium, which is known to protect gut microbial
environment, was found in patients with rectal neoplasms [47].
2. Why the authors stated the following: "the controls may have been more conscious about a healthy
body image or healthy lifestyle than the cases.". Please argue. nother limitation is the self-reported
information (recall bias), moreover, cases are more prone to record more detailed information
compared to control, or might over-estimate the consumption resulting in an over-estimated
association between tea consumption and CRC. moreover, cases might modify their lifestyle because
of the disease. further, authors included only korean population. it means that their results cannot be
generalized to other ethinic group, or it should be done with caution. Please add these considerations.
Authors’ response:
Thank you for your valuable comments. As the reviewer recommended, cases are more prone to
record more detailed information such as dietary habits in certain circumstances [Setia MS,
2016]. However, controls who were recruited from health individuals from the Center for
Cancer Prevention and Detection at National Cancer Center in this study have been known as
having higher education (more than 12 years), doing more regular exercise, having higher
monthly income and having lower unemployment compared to cases, as already shown in many
other studies [Shin A et al., 2014, Park Y et al., 2016, Woo H et al., 2016, Lee J et al., 2018, Lee S
et al., 2019 and Kim J et al., 2019]. Moreover, controls are likely to consume less total energy
intake and lower intake of high-inflammatory diet than cases, which has to be consider that
controls may have been more health-conscious than cases in this study.
Additionally, as the reviewer recommended, we addressed the fourth limitation, which this
study included only the Korean population.
References:
Setia MS. Methodology Series Module 2: Case-control Studies (2016) Indian Journal of Dermatology
61(2):146-151
Shin A, Joo J, Yang HR, Bak J, Park Y, Kim J, et al. Risk prediction model for colorectal cancer: National health
insurance corporation study, Korea (2014) PLoS One 9(2):e88079.
Park Y, Lee J, Oh JH, Shin A, Kim J. Dietary patterns and colorectal cancer risk in a Korean population: A casecontrol study (2016) Medicine (Baltimore) 95(25):e3759.
Woo H, Lee J, Lee J, Park JW, Park S, Kim J, et al. Diabetes mellitus and site-specific colorectal cancer risk in
Korea: a case-control study (2016) Journal of Preventive Medicine and Public Health 49(1):45-52.
Lee J, Shin A, Oh JH, Kim J. The relationship between nut intake and risk of colorectal cancer: a case control
study (2018) Nutrition Journal 17(1):37.
Lee S, Woo H, Lee J, Oh JH, Kim J, Shin A. Cigarette smoking, alcohol consumption, and the risk of colorectal
cancer in South Korea: A case-control study (2019) Alcohol 76:15-21.
Kim J, Lee J, Oh JH, Chang HJ, Sohn DK, Kwon O, et al. Dietary lutein plus zeaxanthin intake and DICER1
rs3742330 A > G polymorphism relative to colorectal cancer risk (2019) Scientific Reports 9(1):3406.
In the manuscript:
Pages 11, Discussion, lines 284-285:
(Lines 284-285) Fourth, because this study included only the Korean population, this result cannot be
generalized; thus, it should be interpreted with caution.

Reviewer 2 Report

There are still some minor problems that needs to be addressed.

For the interaction model, was the tea consumption treated as categorical variable? If that is the case, I would expect two interaction terms and two p values, what does the interaction p value reported in Table 4 represent? Please clarify it in the table footnote.

Table 4 listed colon cancer twice

Line 163 summarize “a decreased risk of ... and a healthy lifestyle, such as a low BMI, increased physical activity and a low-inflammatory diet”. The conclusion about low-inflammatory diet was not supported by this study. Also line 185, the authors stated “high BMI, physical inactivity and an unhealthy diet were independent risk factors”. The unhealthy diet was not supported in by this study.

Throughout the manuscript, the authors kept saying "combined effect of lifestyle and green tea consumption", I had hard time with this "combined effect". "Combined effect" was not interaction. i would suggest the authors frame it as something like "the protective effect of green tea varies by other lifestyle factors" or "green tea interact with other lifestyle factors to influence CRC cancer risk"

Author Response

Responses to Reviewer #2
General comment:
There are still some minor problems that needs to be addressed.
Authors’ response:
We appreciate the reviewer’s valuable comment. We have modified our manuscript
accordingly. Our responses to the 4 comments are provided as follows.
Specific essential comments:
1. For the interaction model, was the tea consumption treated as categorical variable? If that is the
case, I would expect two interaction terms and two p values, what does the interaction p value
reported in Table 4 represent? Please clarify it in the table footnote.
Authors’ response:
Thank you for your valuable comments. The interaction p values reported in Table 4 were
calculated from joint test, which does not exactly have the significance level. Therefore, we have
clarified the previous p values in the footnote of Table 4 and have recalculated the interaction p
values from analysis of maximum likelihood estimates (data not shown). Based on these p
values, we have presented OR and 95% CI of CRC, colon cancer and rectal cancer stratified by
tobacco smoking, alcohol drinking and DII in figure 1, 2 and 3 as follows. For clarification, we
have revised some parts of the Materials and Methods and Result section as follows.
[Related to comment #1-1] Table 4. Protective effect of green tea consumption and health-related factors on CRC
Green tea intake (g/d)
p value 1
T1 (≤0.01) T2 (0.02-25.49) T3 (≥25.50)
CRC
Prior BMI
≥25 kg/m2
1.0 (ref) 1.16 (0.77-1.74) 0.45 (0.28-0.71)
0.36
<25 kg/m2
0.67 (0.48-0.93) 1.17 (0.72-1.92) 1.49 (0.86-2.58)
Physical
activity
No 1.0 (ref) 1.34 (0.99-1.81) 0.59 (0.43-0.82)
0.92
Yes 0.48 (0.35-0.67) 0.91 (0.56-1.45) 0.97 (0.58-1.63)
Smoking
Ever 1.0 (ref) 1.59 (1.17-2.17) 0.55 (0.38-0.78)
0.041
Never 1.23 (0.88-1.72) 0.61 (0.38-0.78) 1.12 (0.67-1.87)
Alcohol
drinking
Ever 1.0 (ref) 1.58 (1.19-2.08) 0.61 (0.45-0.84)
0.024
Never 1.07 (0.75-1.52) 0.51 (0.31-0.84) 0.89 (0.52-1.53)
DII
High (≥2.73) 1.0 (ref) 1.65 (1.21-2.25) 0.60 (0.41-0.85)
0.06
Low (<2.73) 1.05 (0.76-1.46) 0.58 (0.36-0.92) 0.90 (0.54-1.51)
Colon
cancer
Prior BMI
≥25 kg/m2
1.0 (ref) 1.07 (0.66-1.73) 0.48 (0.28-0.81)
0.76
<25 kg/m2
0.53 (0.35-0.79) 1.21 (0.66-2.20) 1.24 (0.63-2.42)
Physical
activity
No 1.0 (ref) 1.34 (0.93-1.95) 0.58 (0.38-0.88)
0.63
Yes 0.58 (0.39-0.88) 0.75 (0.41-1.35) 0.85 (0.43-1.65)
Smoking
Ever 1.0 (ref) 1.49 (1.00-2.21) 0.52 (0.32-0.84)
0.2
Never 1.50 (0.99-2.27) 0.63 (0.35-1.12) 1.06 (0.55-2.03)
Alcohol
drinking
Ever 1.0 (ref) 1.44 (1.01-2.04) 0.59 (0.39-0.88)
0.19
Never 1.06 (0.69-1.64) 0.56 (0.30-1.04) 0.80 (0.40-1.60)
DII
High (≥2.73) 1.0 (ref) 1.45 (0.98-2.14) 0.51 (0.32-0.83)
0.29
Low (<2.73) 1.02 (0.68-1.55) 0.67 (0.37-1.20) 1.05 (0.54-2.04)
Rectal
cancer
Prior BMI
≥25 kg/m2
1.0 (ref) 1.35 (0.80-2.30) 0.38 (0.19-0.74)
0.16
<25 kg/m2
0.80 (0.52-1.25) 1.09 (0.58-2.06) 2.08 (0.96-4.51)
Physical
activity
No 1.0 (ref) 1.41 (0.98-2.03) 0.61 (0.40-0.93)
0.91
Yes 0.39 (0.25-0.62) 1.07 (0.57-1.99) 1.16 (0.58-2.34)
Smoking
Ever 1.0 (ref) 1.77 (1.21-2.59) 0.57 (0.36-0.90)
0.08
Never 1.00 (0.65-1.54) 0.60 (0.33-1.10) 1.25 (0.64-2.44)
Alcohol
drinking
Ever 1.0 (ref) 1.78 (1.25-2.54) 0.67 (0.44-1.00)
0.06
Never 1.14 (0.73-1.80) 0.47 (0.25-0.91) 0.90 (0.44-1.83)
DII
High (≥2.73) 1.0 (ref) 1.99 (1.35-2.94) 0.73 (0.46-1.17)
0.04
Low (<2.73) 1.20 (0.79-1.83) 0.46 (0.25-0.84) 0.69 (0.35-1.35)
BMI, body mass index; DII, dietary inflammatory index; T, tertile. Multivariate models are constructed to calculate OR and 95% CI for the risk of CRC adjusting for first-degree family
history of CRC and education. Mutually adjusted for prior BMI, physical activity, smoking, alcohol drinking and DII, if applicable. 1
p value was calculated from joint tests
In the manuscript:
Pages 3, Materials and Methods, lines 114-116:
(Lines 114-116) To further clarify the risk of CRC depending on the health-related factors, the
association between green tea consumption and CRC risk was stratified by smoking, alcohol
consumption and DII, which may have some interaction in the analysis.
Pages 7, Result, lines 154-163:
(Lines 154-163) 3.4. Protective effect of green tea consumption and health-related factors on CRC
The protective effect of green tea consumption and modifiable lifestyle factors on the risk of CRC is
shown in Table 4. Different and obscured risk patterns were shown depending on the green tea and
lifestyle factors. The association between green tea consumption and the risk of CRC, colon cancer
and rectal cancer further, stratified by tobacco smoking, alcohol consumption and DII are shown in
figure 1, 2 and 3, respectively. Moderate green tea consumption increased the risk of CRC in eversmokers, ever-drinkers and high DII, but heavy tea consumption consistently decreased the risk of
CRC. In never-smokers, never-drinkers and low DII, high consumption has a protective effect, but
moderate green tea consumption has no effect, and these protective effects seemed to be more
relevant for rectal cancer than for colon cancer.
[Related to comment #1-2] Figure 1. Green tea consumption and the risk of CRC stratified by demographic characteristics
[Related to comment #1-3] Figure 2. Green tea consumption and the risk of colon cancer stratified by demographic characteristics
[Related to comment #1-4] Figure 3. Green tea consumption and the risk of rectal cancer stratified by demographic characteristics
2. Table 4 listed colon cancer twice.
Authors’ response:
As the reviewer recommended, we have revised the word colon to rectal in Table 4.
[Related to comment #2-1] Table 4. . Protective effect of green tea consumption and health-related factors on CRC
Green tea intake (g/d)
p value 1
T1 (≤0.01) T2 (0.02-25.49) T3 (≥25.50)
CRC
Prior BMI
≥25 kg/m2
1.0 (ref) 1.16 (0.77-1.74) 0.45 (0.28-0.71)
0.36
<25 kg/m2
0.67 (0.48-0.93) 1.17 (0.72-1.92) 1.49 (0.86-2.58)
Physical
activity
No 1.0 (ref) 1.34 (0.99-1.81) 0.59 (0.43-0.82)
0.92
Yes 0.48 (0.35-0.67) 0.91 (0.56-1.45) 0.97 (0.58-1.63)
Smoking
Ever 1.0 (ref) 1.59 (1.17-2.17) 0.55 (0.38-0.78)
0.041
Never 1.23 (0.88-1.72) 0.61 (0.38-0.78) 1.12 (0.67-1.87)
Alcohol
drinking
Ever 1.0 (ref) 1.58 (1.19-2.08) 0.61 (0.45-0.84)
0.024
Never 1.07 (0.75-1.52) 0.51 (0.31-0.84) 0.89 (0.52-1.53)
DII
High (≥2.73) 1.0 (ref) 1.65 (1.21-2.25) 0.60 (0.41-0.85)
0.06
Low (<2.73) 1.05 (0.76-1.46) 0.58 (0.36-0.92) 0.90 (0.54-1.51)
Colon
cancer
Prior BMI
≥25 kg/m2
1.0 (ref) 1.07 (0.66-1.73) 0.48 (0.28-0.81)
0.76
<25 kg/m2
0.53 (0.35-0.79) 1.21 (0.66-2.20) 1.24 (0.63-2.42)
Physical
activity
No 1.0 (ref) 1.34 (0.93-1.95) 0.58 (0.38-0.88)
0.63
Yes 0.58 (0.39-0.88) 0.75 (0.41-1.35) 0.85 (0.43-1.65)
Smoking
Ever 1.0 (ref) 1.49 (1.00-2.21) 0.52 (0.32-0.84)
0.2
Never 1.50 (0.99-2.27) 0.63 (0.35-1.12) 1.06 (0.55-2.03)
Alcohol
drinking
Ever 1.0 (ref) 1.44 (1.01-2.04) 0.59 (0.39-0.88)
0.19
Never 1.06 (0.69-1.64) 0.56 (0.30-1.04) 0.80 (0.40-1.60)
DII
High (≥2.73) 1.0 (ref) 1.45 (0.98-2.14) 0.51 (0.32-0.83)
0.29
Low (<2.73) 1.02 (0.68-1.55) 0.67 (0.37-1.20) 1.05 (0.54-2.04)
Rectal
cancer
Prior BMI
≥25 kg/m2
1.0 (ref) 1.35 (0.80-2.30) 0.38 (0.19-0.74)
0.16
<25 kg/m2
0.80 (0.52-1.25) 1.09 (0.58-2.06) 2.08 (0.96-4.51)
Physical
activity
No 1.0 (ref) 1.41 (0.98-2.03) 0.61 (0.40-0.93)
0.91
Yes 0.39 (0.25-0.62) 1.07 (0.57-1.99) 1.16 (0.58-2.34)
Smoking
Ever 1.0 (ref) 1.77 (1.21-2.59) 0.57 (0.36-0.90)
0.08
Never 1.00 (0.65-1.54) 0.60 (0.33-1.10) 1.25 (0.64-2.44)
Alcohol
drinking
Ever 1.0 (ref) 1.78 (1.25-2.54) 0.67 (0.44-1.00)
0.06
Never 1.14 (0.73-1.80) 0.47 (0.25-0.91) 0.90 (0.44-1.83)
DII
High (≥2.73) 1.0 (ref) 1.99 (1.35-2.94) 0.73 (0.46-1.17)
0.04
Low (<2.73) 1.20 (0.79-1.83) 0.46 (0.25-0.84) 0.69 (0.35-1.35)
BMI, body mass index; DII, dietary inflammatory index; T, tertile. Multivariate models are constructed to calculate OR and 95% CI for the risk of CRC adjusting for first-degree family
history of CRC and education. Mutually adjusted for prior BMI, physical activity, smoking, alcohol drinking and DII, if applicable. 1
p value was calculated from joint tests
3. Line 163 summarize “a decreased risk of ... and a healthy lifestyle, such as a low BMI, increased
physical activity and a low-inflammatory diet”. The conclusion about low-inflammatory diet was not
supported by this study. Also line 185, the authors stated “high BMI, physical inactivity and an
unhealthy diet were independent risk factors”. The unhealthy diet was not supported in by this study.
Authors’ response:
Thank you for your sincere comments. As the reviewer recommended, we have deleted ‘the
unhealthy diet’. Instead, we have revised the Discussion section considering three figures
attached above as follows.
In the manuscript:
Pages 9, Discussion, lines 176-181:
(Lines 176-181) Overall, high green tea consumption, with or without considering lifestyle factors,
decreased the risk of CRC. However, regarding moderate green tea intake, different risk patterns were
observed depending on healthy and unhealthy lifestyles, especially the increased risk of CRC was
shown among ever-smokers, ever-drinkers and high-inflammatory diet group. The protective effect of
green tea on the risk of CRC varied by lifestyle factors was likely to be stronger in rectal cancer
patients than in colon cancer patients.
Pages 9, Discussion, lines 201:
(Lines 201) High BMI and physical inactivity were independent risk factors for CRC in this study…
Pages 10, Discussion, lines 213-216:
(Lines 213-216) We observed significant interactions green tea consumption with tobacco smoking,
alcohol drinking and high DII (data not shown). Moreover, heavy green tea intake was associated with
a decreased risk of CRC, but inconsistent risk was found depending on lifestyle factors among
moderate intake of green tea group…
Pages 10, Discussion, lines 219-224:
(Lines 219-224) we found different risk patterns for CRC by the amount of green tea consumption
depending on smoking and drinking status and high-inflammatory diet. Heavy green tea consumption
decreased the risk of CRC, but moderate green tea consumption increased the risk of CRC in eversmokers, ever-drinkers and high DII groups. In never-smokers, never-drinkers and low DII, high
green tea consumption also has a protective effect, but moderate consumption has no effect.
Pages 10, Discussion, lines 230-233:
(Lines 230-233) First, each lifestyle factor, smoking, alcohol consumption and high-inflammatory
diet, might attenuate the protective effect of green tea on CRC. It is clear that the examined factors are
known to be associated with predispositions to various diseases, including cancer [22, 35, 36].
4. Throughout the manuscript, the authors kept saying "combined effect of lifestyle and green tea
consumption", I had hard time with this "combined effect". "Combined effect" was not interaction. i
would suggest the authors frame it as something like "the protective effect of green tea varies by other
lifestyle factors" or "green tea interact with other lifestyle factors to influence CRC cancer risk"
Authors’ response:
Based on the reviewer’s recommendation, we have revised all throughout the manuscript.
In the manuscript:
Pages 1, Title, lines 2-3:
(Lines 2-3) Protective effect of green tea consumption varies by lifestyle factors on colorectal cancer
Pages 1, Abstract, lines 19:
(Lines 19) CRC was altered by the protective effect of green tea…
Pages 2, Introduction, lines 49:
(Lines 49) the protective effect of green tea intake interacting with lifestyle factors on CRC risk in
South…
Pages 2, Introduction, lines 54:
(Lines 54) was altered by the interaction between green tea consumption and lifestyle factors…
Pages 3, Materials and Methods, lines 112:
(Lines 112) interaction between lifestyle factors and green tea consumption…
Pages 7, Result, lines 154-163:
(Lines 154-163) 3.4. Protective effect of green tea consumption and health-related factors on CRC
The protective effect of green tea consumption and modifiable lifestyle factors on the risk of CRC is
shown in Table 4. Different and obscured risk patterns were shown depending on the green tea and
lifestyle factors. The association between green tea consumption and the risk of CRC, colon cancer
and rectal cancer further, stratified by tobacco smoking, alcohol consumption and DII are shown in
figure 1, 2 and 3, respectively. Moderate green tea consumption increased the risk of CRC in eversmokers, ever-drinkers and high DII, but heavy tea consumption consistently decreased the risk of
CRC. In never-smokers, never-drinkers and low DII, high consumption has a protective effect, but
moderate green tea consumption has no effect, and these protective effects seemed to be more
relevant for rectal cancer than for colon cancer.
[Related to comment #4-1] Table 4. . Protective effect of green tea consumption and health-related factors on CRC
Green tea intake (g/d)
p value 1
T1 (≤0.01) T2 (0.02-25.49) T3 (≥25.50)
CRC
Prior BMI
≥25 kg/m2
1.0 (ref) 1.16 (0.77-1.74) 0.45 (0.28-0.71)
0.36
<25 kg/m2
0.67 (0.48-0.93) 1.17 (0.72-1.92) 1.49 (0.86-2.58)
Physical
activity
No 1.0 (ref) 1.34 (0.99-1.81) 0.59 (0.43-0.82)
0.92
Yes 0.48 (0.35-0.67) 0.91 (0.56-1.45) 0.97 (0.58-1.63)
Smoking
Ever 1.0 (ref) 1.59 (1.17-2.17) 0.55 (0.38-0.78)
0.041
Never 1.23 (0.88-1.72) 0.61 (0.38-0.78) 1.12 (0.67-1.87)
Alcohol
drinking
Ever 1.0 (ref) 1.58 (1.19-2.08) 0.61 (0.45-0.84)
0.024
Never 1.07 (0.75-1.52) 0.51 (0.31-0.84) 0.89 (0.52-1.53)
DII
High (≥2.73) 1.0 (ref) 1.65 (1.21-2.25) 0.60 (0.41-0.85)
0.06
Low (<2.73) 1.05 (0.76-1.46) 0.58 (0.36-0.92) 0.90 (0.54-1.51)
Colon
cancer
Prior BMI
≥25 kg/m2
1.0 (ref) 1.07 (0.66-1.73) 0.48 (0.28-0.81)
0.76
<25 kg/m2
0.53 (0.35-0.79) 1.21 (0.66-2.20) 1.24 (0.63-2.42)
Physical
activity
No 1.0 (ref) 1.34 (0.93-1.95) 0.58 (0.38-0.88)
0.63
Yes 0.58 (0.39-0.88) 0.75 (0.41-1.35) 0.85 (0.43-1.65)
Smoking
Ever 1.0 (ref) 1.49 (1.00-2.21) 0.52 (0.32-0.84)
0.2
Never 1.50 (0.99-2.27) 0.63 (0.35-1.12) 1.06 (0.55-2.03)
Alcohol
drinking
Ever 1.0 (ref) 1.44 (1.01-2.04) 0.59 (0.39-0.88)
0.19
Never 1.06 (0.69-1.64) 0.56 (0.30-1.04) 0.80 (0.40-1.60)
DII
High (≥2.73) 1.0 (ref) 1.45 (0.98-2.14) 0.51 (0.32-0.83)
0.29
Low (<2.73) 1.02 (0.68-1.55) 0.67 (0.37-1.20) 1.05 (0.54-2.04)
Rectal
cancer
Prior BMI
≥25 kg/m2
1.0 (ref) 1.35 (0.80-2.30) 0.38 (0.19-0.74)
0.16
<25 kg/m2
0.80 (0.52-1.25) 1.09 (0.58-2.06) 2.08 (0.96-4.51)
Physical
activity
No 1.0 (ref) 1.41 (0.98-2.03) 0.61 (0.40-0.93)
0.91
Yes 0.39 (0.25-0.62) 1.07 (0.57-1.99) 1.16 (0.58-2.34)
Smoking
Ever 1.0 (ref) 1.77 (1.21-2.59) 0.57 (0.36-0.90)
0.08
Never 1.00 (0.65-1.54) 0.60 (0.33-1.10) 1.25 (0.64-2.44)
Alcohol
drinking
Ever 1.0 (ref) 1.78 (1.25-2.54) 0.67 (0.44-1.00)
0.06
Never 1.14 (0.73-1.80) 0.47 (0.25-0.91) 0.90 (0.44-1.83)
DII
High (≥2.73) 1.0 (ref) 1.99 (1.35-2.94) 0.73 (0.46-1.17)
0.04
Low (<2.73) 1.20 (0.79-1.83) 0.46 (0.25-0.84) 0.69 (0.35-1.35)
BMI, body mass index; DII, dietary inflammatory index; T, tertile. Multivariate models are constructed to calculate OR and 95% CI for the risk of CRC adjusting for first-degree family
history of CRC and education. Mutually adjusted for prior BMI, physical activity, smoking, alcohol drinking and DII, if applicable. 1
p value was calculated from joint tests
Pages 9, Discussion, lines 180-181:
(Lines 180-181) The protective effect of green tea on the risk of CRC varied by lifestyle factors was
likely to be stronger in rectal cancer patients than in colon cancer patients.
Pages 10, Discussion, lines 213-215:
(Lines 213-215) We observed significant interactions green tea consumption with tobacco smoking,
alcohol drinking and high DII (data not shown). Moreover, heavy green tea intake was associated with
a decreased risk of CRC…
Pages 10, Discussion, lines 228:
(Lines 228) evidence on the protective effect of green tea intake by…
Pages 10, Discussion, lines 245:
(Lines 245) The underlying mechanism of the protective effect of green tea depending on lifestyle
factors…
Pages 11, Discussion, lines 263-264:
(Lines 263-264) …may affect the protective effect of green tea depending on lifestyle factors. In the
present study, a similar pattern of protective effect of green tea and lifestyle factors….
Pages 11, Discussion, lines 287-288:
(Lines 287-288) The protective effect of green tea consumption on CRC depending on lifestyle factors
was shown evidently, especially in heavy green tea consumption.
